# Objective Methods of Muscle Tone Diagnosis and Their Application—A Critical Review

**DOI:** 10.3390/s23167189

**Published:** 2023-08-15

**Authors:** Barbora Kopecká, David Ravnik, Karel Jelen, Václav Bittner

**Affiliations:** 1Faculty of Physical Education and Sport, Charles University, 162 52 Prague, Czech Republic; 2Faculty of Health Sciences, University of Primorska, 6310 Izola, Slovenia; 3Faculty of Science, Humanities and Education, Technical University of Liberec, 461 17 Liberec, Czech Republic

**Keywords:** soft tissue mechanical properties, muscle tone, myotonometry, in vivo

## Abstract

“Muscle tone” is a clinically important and widely used term and palpation is a crucial skill for its diagnosis. However, the term is defined rather vaguely, and palpation is not measurable objectively. Therefore, several methods have been developed to measure muscle tone objectively, in terms of biomechanical properties of the muscle. This article aims to summarize these approaches. Through database searches, we identified those studies related to objective muscle tone measurement in vivo, in situ. Based on them, we described existing methods and devices and compared their reliability. Furthermore, we presented an extensive list of the use of these methods in different fields of research. Although it is believed by some authors that palpation cannot be replaced by a mechanical device, several methods have already proved their utility in muscle biomechanical property diagnosis. There appear to be two issues preventing wider usage of these objective methods in clinical practice. Firstly, a high variability of their reliability, and secondly, a lack of valid mathematical models that would provide the observed mechanical characteristics with a clear physical significance and allow the results to be compared with each other.

## 1. Introduction

Palpation is one of the oldest examination techniques. It plays an essential role in the diagnosis of soft tissues of the musculoskeletal system, being one of the basic skills of a physiotherapist [1]. Its use is extensive, including the assessment of the consistency, smoothness, displacement, or extensibility of the examined tissue or the temperature and moisture of the body surface. The painfulness of the studied area can also be estimated based on the patient’s reaction.

In the tradition of the so-called Prague School of Rehabilitation, slow, deep, layered palpation with fingertip pads is used to examine the soft tissues of the musculoskeletal system while the patient is completely relaxed [2]. The underlying structures (skin, subcutaneous tissue, adipose tissue, fascia, muscles, etc.) are compressed by creating sufficient pressure on the body surface. Depending on the depth of palpation, the affected tissue layers’ elasticity and resistance to deformation can be assessed. However, the pressure must not be disproportionately large; otherwise, the examiners perceive the mechanical processes on their fingers and not the tissue under examination [3]. Véle [4] recommends palpation at an angle of 60–90° with the surface. This method is often used to diagnose functional disorders of the soft tissues of the musculoskeletal system, especially the so-called muscle tone. Another approach to assess the mechanical properties of skeletal muscle is the so-called tapotement, or tapping perpendicular to the muscle, where the propagation of the wave in the muscle is observed [1]. In order to investigate trigger points, the so-called strumming is used, which elicits a characteristic response in the form of muscle twitching and referred pain [5].

The palpation techniques described above are widely used, but unfortunately, they are also subjective examination methods. This article aims to summarize the possible approaches to objectify the mechanical properties of soft tissues of the human musculoskeletal system, and to provide a selection of research areas in which these objective methods can be used. The basic questions addressed in the text are as follows: What methods can be used to assess the mechanical properties of skeletal muscle in vivo objectively? How can these methods support subjective palpation examinations? In what research areas have they already been used?

## 2. Muscle Tone

More than a century ago, Foster defined muscle tone as “resistance to passive movements” [6] and this definition is still widely used today [3,7,8,9], serving as the principal criterion for its assessment in clinical practice. Multiple authors have since investigated the physiological nature and components of this phenomenon, inferring that it can be understood as a set of mechanical (rheological) and neurophysiological properties of the skeletal muscle [10,11].

Simons and Mense [11] approach the concept based on the presence of detectable electrical activity in the muscle. That is, based on the detectable EMG (electromyography) signal response of the skeletal muscle. The EMG activity is a manifestation of the excitation of alpha motoneurons, i.e., the tone produced during voluntary but also involuntary (e.g., spasticity, rigidity, spasm) contractions and also during incomplete muscle relaxation (i.e., the measured individual has been instructed to relax but is not able to do so completely). In this case, the muscle’s stiffness is due to the interaction between actin and myosin.

In addition, there is a component of muscle tone unaffected by alpha activity (i.e., EMG-silent), caused by the inherent viscoelastic properties of the tissue. Masi and Hannon [9] describe it as the human resting muscle tone and attribute about 1% of maximal voluntary contraction to it. In addition to the intrinsic mechanical resistance of the muscle, a certain role is attributed to the mutual binding and movement between actin and myosin [9,11]. The latter is also a possible explanation for thixotropy, i.e., the gradual decrease in resistance to movement concerning the preceding movement [11].

Another passive (EMG-silent) component is the stiffness of the surrounding soft tissues, especially fasciae, which also show thixotropy. Muscle contracture due to prolonged stay in a shortened position is a well-known issue, but the inherent variability in the viscoelastic properties of these tissues has not yet been clearly explained. Control by the autonomic nervous system is thought to regulate pressure in small local vessels and plasma washout into the interstitium based on information from receptors, which will affect overall viscosity. At the same time, smooth muscle in the fascia has been demonstrated, implying that the fascia may have a system of tone adjustment, independent of muscle tone. [12] However, when assessing biomechanical properties by palpation or indentation testing, these two components are examined simultaneously. Therefore, some authors [9] suggest using the term “myofascial tone” instead when assessing it.

In addition, there is also contractile muscle activity without electrical activity (in some genetic disorders associated with impaired calcium metabolism in the sarcoplasmic reticulum), which is also the case for myofascial trigger points [11]. Although some studies have shown low EMG activity levels at the trigger point (so-called endplate noise), action potentials are absent in the surrounding rigid core. At the same time, the nature of this particular kind of local hypertonicity is more or less explicitly described in the literature from a physiological point of view. It is assumed that due to the overloading of the muscle, there is a malfunction in the muscle endplate, which continuously flushes out acetylcholine and thus causes a sustained contraction at the site. This contraction is energy depleting; Simons [13] described this as a local energy crisis. Therefore, there is a lack of energy for the reuptake of the sarcoplasmic reticulum, which further promotes contraction. The local ischemia also causes typical soreness.

Partly outside the categories of Simons and Mense [11] lies muscle tone as a measure of “alertness”, i.e., the readiness of a muscle to contract. Latash and Zatsiorski [14] define it as such and describe it as the threshold for eliciting the stretch reflex. This threshold is set from the central nervous system (e.g., by a conscious instruction that the muscle is to be relaxed), and an increase in this threshold may manifest itself, for example, in the inability to achieve a silent EMG when attempting to relax. At the same time, however, it is generally accepted that the gamma system sets muscle excitability through the muscle spindles and from different levels of the central nervous system [3,4,15,16,17]. There is as yet no evidence of EMG manifestations in this context.

Given the above, it is clear that muscle tone is a very complex term and, despite being commonly used in professional literature, it lacks a precise definition [14] and is still a subject of discussion and research, see, e.g., [3,7,8,9,11]. Defining muscle tone as resistance to passive movement is very vague, allowing for ambiguous interpretations [18,19]. Furthermore, the resistance is influenced by other factors besides the muscle and its fascia, such as the joint range of motion, ligament laxity, etc. [3,11,14]. Moreover, while Foster describes it as “independent of all distinct muscular contractions of volitional or other origin” [6], other authors assume that the muscle tone actively contributes to movement [20] and some even differentiate between passive and active muscle tone [9], countering the supposition that the patient has to be completely relaxed for the measurement.

Variability of views and the absence of an explicit definition of muscle tone make it necessary to always specify in what sense this term is understood when speaking of examination of muscle tone. In clinical practice, the healthcare practitioners commonly examine it using palpation, hence the muscle tone in this context refers to the examiner’s perception, i.e., the resistance of the tissue against the palpating finger. Curiously, this criterion for the assessment of muscle tone is cited much less frequently then the already mentioned resistance to passive movement [3,4,9,14].

The expected level of the resistance is not clearly established [21], however, therapists commonly compare this resistance with experience from previous practice. Furthermore, the resistance can be compared, for example, between muscles of one individual or, in the case of long-term therapeutic care, changes in one muscle (site) over time. Above all, however, it is possible to compare one particular muscle (site) before and after therapeutic intervention. Intra-individual comparisons also minimize the influence of the mechanical properties of the surface tissues (skin, fat layer, etc.) on the result, as these are likely to vary only minimally between the individual examinations being compared. It assumes that there is no significant change in body composition due to dehydration, reduction in body fat, etc. Furthermore, if the patient is asked to relax during the test, the voluntary instruction from the central nervous system can be expected to be the same for both examinations being compared. The patient’s ability to relax can then be attributed to the effectiveness of the intervention and should not be considered a confounding factor of the test.

Note that this overview of muscle tone definitions is only a brief summary, serving to illustrate how we approach the term within this paper and why. For a more comprehensive review on muscle tone, see for example [9,14,19].

## 3. Methods

The preliminary search of information on the available objective methods for the diagnosis of muscle tone was based on the long-term specialization and research of the authors and their extensive research of the literature available. Regarding the application and reliability of the methods described, the relevant studies were identified mostly through searching electronic databases. The principal database used was PubMed. Additionally, the clinical application of Myoton was also documented using references listed on the official Myoton website. Only studies written in English were considered. The search was conducted on several occasions, the last being on 28 July 2023. A simplified PRISMA diagram [22] describing the review process of the reliability of the most frequently used methods is shown in Figure 1. The keywords used were ICC/s, healthy, and muscle in combination with of the following: CMT, MRE, myotonometer, myotonometry, MyotonPRO, SSI, or SWE. Only studies that were available in full text, tested the reliability on skeletal muscle in vivo and in a healthy population aged 18–60, and provided the ICC as an indicator of the method’s reliability were included. Three of the included studies tested multiple methods.

The keywords for application included muscle tone, mechanical properties, elasticity/stiffness, myotonometry, elastography, and similar. In order to be included in the review, the objective method described/used had to be “in vivo, in situ” and performed on skeletal muscles. The variability of clinical application of the described methods proved to be wide and extensive, so the list is not exhaustive.

## 4. Objective Methods of Muscle Tone Diagnosis

Latash and Zatsiorski [14] defined three possible approaches to the concept of muscle tone, from which the respective methods of its objectification proceed. The first one is based on the definition of tone as the resistance of a muscle to movement at a given joint. This approach was used by McPherson [8] and Brennan [23] in their works and proposed devices. They used the resistive force of the spring that kept the segment of the respective musculoskeletal system out of the physiological position in a given joint to determine the degree of spasticity. The very same principle is the basis of the widely used subjective assessment of spasticity based on the Ashworth scale or its modified version [3,24,25]. However, these methods assess the mechanical properties of the entire musculoskeletal chain, including the joint. Thus, they are not focused on a single muscle and small changes in muscle tone and the mechanical properties of a particular muscle (elasticity, viscosity) are difficult to determine from them, see [10,26].

The second approach is to use an EMG, where muscle tone is taken as the initial resting signal level without muscle activation. This is the approach taken by Jacobson [27], but as views on muscle tone have changed, this method has proven inadequate: Adrian and Bronk [28] found that completely relaxed normal muscle shows no spontaneous electrical activity, but under such conditions, quantifiable muscle tone (in terms of hypo-, eu-, or hypertonia) can still be detected. Latash and Zatsiorski [14] also point out that complete relaxation might not be possible in certain patients or for specific measurement scenarios.

The third group of methods are indentation stress tests. Their principle is to press a tip (indenter) of known geometry into a body surface and monitor the mechanical response, or the load characteristics, of the underlying tissue. Two types of such devices, often called myotonometers, can be distinguished. In the first case, the oscillatory response of the tissue to a single, short, rectangular pulse of the indenter is evaluated. Probably the most well-known representative of these devices is the Myoton, or the latest model MyotonPRO [29,30]. The reliability of the method varies between 0.74–0.93 (95% CI_Avg_, see Table 1), in different measurements [31,32,33,34,35,36,37,38,39,40,41,42,43,44,45]. Its application is so far limited to superficial muscles and other soft tissues. It cannot examine deep, hard-to-palpate muscles and cannot measure thin or small muscles. A major shortcoming of the device is that it uses tapping, an impulse different from normal palpation, which can elicit an unwanted reflex response in the tissue that can distort the result.

The second subgroup consists of myotonometers whose indenter pushes against the tissue to a predefined depth or to a predefined resistance force at a defined but relatively low speed, and then it returns. Such devices essentially simulate the palpation technique described above. The output is the dependence of the resistive force of the tissue on the indentation depth of the indenter and takes the form of a hysteresis curve. The first devices of this type were originally developed to determine the pressure required to induce pain in soft tissue [46]; only later was a measurement of pliability added [47], but this was not very reliable [48]. Gradually, different authors [49,50,51,52,53,54,55,56] worked on developing more accurate and user-friendly solutions. For some of these first devices, it was necessary to perform the indentation manually, which did not allow a constant force or speed of tip indentation to be ensured [49]. Consequently, some authors developed manual instruments that perform the indentation and calculation of rheological properties automatically, but the necessity of manual stabilization causes inaccuracies in the obtained results. For example, Leonard et al.’s Myotonometer [52], which shows variable reliability between 0.42 and 1.00 when used on selected muscles in different experimental settings [57,58,59], is worth mentioning. Other instruments [55,56,60] have a fixed design that minimizes the unreliability associated with examiner participation at the cost of less user friendliness. For example, the *CMT* (computerized muscle tonometer) has been found to have a reliability between 0.82 and 0.97 (95% CI_Avg_, see Table 1) [55,61]. Another option is to attach the device directly to the segment to be measured. In combination with constant tissue deformation by the indenter, this option is used by the MC Sensor [62].

A general criticism of Latash and Zatsiorski [14] against indentation methods is that they treat tone as a passive property only. Thus, they do not consider its participation in active movement and posturing since the examinee is always instructed to relax. However, it can be argued that, in manual palpation, complete relaxation is a requirement for correct examination as well according to some authors [2], and even Latash and Zatsiorski [14] admit that muscle tone can be defined based on the state when the muscle is “relaxed to the maximum of the examinee’s abilities”.

From a physical point of view, it is important to note that the measurement of the load characteristic alone only allows a limited assessment of the mechanical properties of the tissues under examination. It is usually based on a descriptive characterization of the tissue response curve as a function of its load. Only by using a suitable mathematical model can the viscoelastic properties of the indented tissue layers be identified. However, the validation of these models is quite complicated and often tied to computational simulations.

In addition to indentation stress tests, there are efforts to determine the mechanical properties of musculoskeletal soft tissues using free vibration techniques (e.g., [63]). Again, information on the rheological properties of the affected muscle tissue can be extracted from these using a suitable mathematical model. However, there are once more limitations associated with the validation of mathematical models, see above.

An alternative to strain stress testing, standing outside of Latash and Zatsiorski’s [14] categories and currently coming to the fore, is non-invasive imaging. Ultrasonic elastography [64,65,66], where the propagation of acoustic waves is sensed by ultrasound, has a relatively long history and widespread use. There are already many approaches for the use of ultrasound in the assessment of tissue rheological properties [67], the most widely used being the supersonic shear wave imaging (SSI) [68]. The method has so far been attributed with a variable reliability ranging between 0.67 and 0.92 (95% CI_Avg_, see Table 1) by various studies [33,40,41,69,70,71,72,73,74,75,76,77,78,79,80,81].

Magnetic resonance elastography (MRE) is a more costly and time consuming but also more accurate option [82,83,84,85]. Magnetic resonance imaging captures the propagation of mechanical waves in the tissue, and stiffness can then be inferred from both the wavelength and the speed of propagation. Low [86] aptly called this technique “virtual palpation.” MRE can even be used for measurement under dynamic conditions in real time [87]. The most advanced technique uses the 3T magnetic field, with a reliability of 0.65–0.98 (95% CI_Avg_, see Table 1) on selected muscles [88,89].

Given the high variability of the reliability indices, a variance decomposition was used for selected methods (MyotonPRO, CMT, SSI). Results (see Table 1) show that the intra-class variability of the reliability indices is significantly higher than the inter-class variability in all the analyzed methods. This virtually makes it impossible to compare the results from different measurements, even if the objective method used was the same.

In order to supplement this, infrared thermography can be mentioned as an indirect method of muscle tone assessment offered by Maršáková and Nováková [90]. In this technique, the body surface temperature is compared with the palpation findings, and the method can only be described as indicative.

**Table 1 sensors-23-07189-t001:** Comparison of the reliability of the objective methods for muscle tone diagnosis.

	MyotonPRO	CMT	SWE (SSI)	MRE
Number of studies included	15	2	16	2
Number of ICC indexes included	240	10	123	14
Diagnosed muscles:				
Abductor Digiti Minimi			X	
Along spine (not specified)			X	
Biceps Brachii	X		X	
Biceps Femoris	X		X	
Deltoideus Anterior	X			
Diaphragma			X	
Erector Spinae	X		X	
Extensor Carpi Radialis Brevis	X			
Flexor Carpi Ulnaris	X			
Gastrocnemius Medialis	X		X	
Gastrocnemius Lateralis	X		X	
Gluteus Maximus			X	
Infraspinatus	X		X	X
Longissimus Thoracis	X			
Masseter	X		X	
Multifidus	X		X	
Rectus Femoris	X	X	X	
Soleus	X			
Splenius Capitis	X			
Supraspinatus				X
Tibialis Anterior	X			
Trapezius				X
Vastus Medialis	X		X	
Vastus Lateralis	X	X	X	
ICC_Avg_ (95% CI)	0.86 (0.84–0.88)	0.90 (0.87–0.93)	0.83 (0.80–0.85)	0.90 (0.85–0.95)
ICCmin–ICCmax	0.06–1.00	0.75–0.99	0.08–1.00	0.16–1.00
95% CI_Avg_	0.74–0.93	0.82–0.97	0.67–0.92	0.65–0.98
Intergroup variability (%)	15	29	17	22
Intragroup variability (%)	85	71	83	78
Studies included	[31,32,33,34,35,36,37,38,39,40,41,42,43,44,45,55,61,69,70,71,72,73,74,75,76,77,78,79,80,81,88,89]			
ICC	Intraclass Correlation Coefficient			
ICC_Avg_	Average ICC of all included studies			
CI	Confidence Interval			
95% CI_Avg_	Average 95% confidence interval for ICC of all included studies			
CMT	Computerized Muscle Tonometer			
SWE	Shear–Wave Elastography			
SSI	Supersonic Shear Imaging			
MRE	Magnetic Resonance Elastography			

## 5. Application of Objective Methods in Muscle Tone Diagnosis

Despite the shortcomings described above, objective methods offer an important advantage over subjective methods. While palpation can only assess muscle tone in terms of several levels (hypotonic/eutonic/hypertonic muscle), objective methods provide a numeric value. Therefore, provided that an appropriate methodology is followed, they allow quantification of intra- and possibly inter-individual differences in the measured components of muscle tone. This allows implementation of quantitative methods into a wide range of research areas. In the following subsections, we present a brief selection of them.

### 5.1. Physiotherapy

In physiotherapy and rehabilitation medicine, or even neurology and other related fields, objective methods can play an important role in assessing muscle tone. First and foremost, objective muscle tone data can help the health care professional determine the lesion or other deviation from the physiological norm, its extent and degree, and possibly its nature or cause. For example, using both USE (ultrasound elastography) [91] and MRE, it has been possible to detect changes in various myopathies [92], conditions preceding pressure ulcers [93], and other muscle pathologies. Furthermore, these methods or Myoton can be used to detect stiff fascicles characteristic of myofascial trigger points [66,94,95], to assess rigidity [96,97,98,99,100] or spasticity [101]. Furthermore, Myoton, MRE, or USE have been used to investigate how changes in the rheological properties of the muscle correlate with the occurrence of various pain [102,103], changes in mobility and position of body segments [104,105,106] or previous injuries [107,108], how masticatory muscle stiffness affects masticatory abilities [109], etc.

Secondly, these methods can supplement the missing link in assessing the effectiveness of various physiotherapy interventions, such as techniques targeted at influencing muscle tone (ultrasound, soft tissue techniques, Kinesio taping, post-isometric relaxation, etc.), but also techniques targeted at another problem (symptom) or holistic techniques in which the change in muscle tone is a secondary manifestation (mobilization techniques, techniques based on neurophysiology, etc.). Similarly, of course, they can be used outside physiotherapy itself, e.g., to assess the effectiveness of medication, surgical interventions, etc. This use can contribute not only to the general validation of methods according to the rules of evidence-based medicine, but also in clinical practice for the assessment of individually implemented interventions on specific patients. For example, MRE has already been used to assess the effect of eccentric exercise [110] or positive thermotherapy [111], USE in instrumental massage [112] and botulinum toxin application [113], Myoton to assess the effectiveness of petrissage (deep muscle massage) and lymphatic drainage [114] and ischemic compression of myofascial trigger points [115], neurodynamics and instrumental soft tissue mobilization [116], strengthening and stretching exercises [117], aquatic exercise and electrical neuromodulation [118], dry needling [119], botulinum toxin and shock wave application [120], mobilization [121], and many other therapeutic modalities.

Thirdly, these methods make it possible to assess and recognize some negative influences on muscle tone objectively. For example, Myoton has been used to determine the effect of army boots on the stiffness of the lower limb muscles [122] and the effect of dental protectors on the stiffness of masticatory muscles [123]. This use in ergonomics is a separate chapter (see below).

### 5.2. Ergonomics

Since occupational therapy is closely related to physiotherapy, the use of these methods is, to some extent, identical in these areas. Objective methods of muscle tone assessment can be used to determine the extent of impairment [66,94,95,96,97,98,99,100,101,124], which can then help in designing appropriate therapy and possible compensatory aids.

They can also be used to assess the effect of a specific workload or work environment on the musculoskeletal system, either by comparing the results of tone measurements on specific muscles before and after working hours within a single day [125] or periodically over a longer time cycle [126], or by comparing measurements under normal circumstances and under specific working conditions [127]. At the same time, it is possible to investigate how the rheological properties found in response to work/stress correlate with factors such as age, duration of employment [126], as well as perceived pain [125].

Similarly, these methods can be used to assess the effectiveness of various ergonomic devices and measures to make work easier for workers or to minimize the adverse effects of their work on their health, especially the musculoskeletal system [128,129,130].

### 5.3. Sport

In the field of sports, objective methods of muscle tone assessment make it possible to investigate the influence of individual types of loads or even specific sports on the muscular apparatus or the rheological properties of the muscle. Myoton has been used for this purpose in many cases, either to measure the immediate effect (i.e., before and after exercise) [131,132,133,134] or to determine the long-term effect. The latter has been determined either by measuring it in a single individual before, during, and after a training program [135], or by simply comparing values in a specific group of athletes with the general population [136,137]. These measurements can give us information on how to enhance performance in some cases and in which cases muscle overload occurs. Thus, this information can be used to prevent injuries and chronic overuse, modify training and recovery methods, and so on. Similarly, the influence of various sports aids, such as the aforementioned dental protectors, can also be investigated [123]. Secondarily, how tone is influenced by other factors such as posture and positioning in different segments can be investigated, and how these may relate to pain [104].

Furthermore, these methods can be used to investigate how the rheological properties of skeletal muscles are related to specific sports performance. For example, measurements of pre-exercise muscle tone using Myoton have shown that, for some muscles, higher agonist muscle tone (or lower antagonist muscle tone [138]) predicts better performance both between different individuals [139,140,141] and within the same individual in the course of a day [142].

Finally, as in physiotherapy, these methods can be applied in sports to test the effectiveness of recovery procedures intended to affect muscle tone or to speed up the treatment of sports injuries. In addition to the cases in the Physiotherapy section and many others, we can mention, for example, the use of negative ion patches in the field of alternative medicine [143].

Apart from one exception [137], we could not find any cases in which a device other than Myoton was used in sports. It can be assumed that this is mainly due to practicality, time and money savings, and availability of the method. Imaging elastography methods, especially MRE, are more likely to be available at healthcare workplaces, whereas sports research is most often performed in sports institutions and laboratories or directly at sports venues.

### 5.4. Basic Research

Methods of muscle tone objectification can also contribute to research on the nature of muscle tone and its behavior in specific physiological and pathological circumstances. Some authors have already used USE [144], MRE [145,146,147], or Myoton [106,109,142,148,149] to establish normative values and to investigate changes in muscle rheological properties during the day or as a function of age, gender, menstrual cycle phase, BMI, race, individual physical activity, or stride length. As mentioned above, Myoton has also served to investigate the effect of specific physical activities on the rheological properties of muscle [131,135,136,137] or how these characteristics further relate to endurance and contractile ability and muscle strength [138,140].

Others have used these methods to investigate the rheological properties of muscles and the nature of their changes in pathologies such as hyperthyroidism [150], myopathy [91,151], other neuromuscular disorders [152,153], changes due to irradiation of tumors [154], etc., but also under specific extreme conditions [127].

## 6. Discussion

There seem to be several obstacles to attempts to objectify assessment of muscle tone. Firstly, it is the ambiguity of the term muscle tone (see Section 3). In this context it is worth mentioning that some authors, e.g., [3,155], believe palpation cannot be replaced by any instrument. Their argument is that the examiner not only applies static pressure but exerts subtle and purposeful movements and thus registers a combination of several pieces of information using receptors for touch, pressure, movement, and position. It must also be acknowledged that the devices mentioned above are not capable of recording the feedback, the patient’s response, that a particular purposeful movement elicits. On the other hand, the disadvantage of proprioception and palpation is that it is not reproducible, and its interpretation is purely subjective. In addition, the experience of the therapist is essential for the quality of the palpation examination. However, it must be acknowledged that the reliability and validity of objective methods depends on examiner’s experience as well, among other factors.

Furthermore, it must be considered that instrumental methods for diagnosing the mechanical properties of skeletal muscle are not intended to replace palpation. Instead, their development is driven by the desire to objectify one of the modalities that can be detected from palpation, since the therapist also inserts a finger against soft tissues, including muscle, and registers their resistance. This palpation technique is described by Kolář [3] as “applying mere pressure.” The instrumental methods can therefore support the palpation examination by objectively quantifying the resistance of the tissue.

Another obstacle that limits the broader deployment of these techniques in clinical practice is the high variability of their reliability. A separate problem, particularly relevant to indentation methods, is the frequent absence of a valid mathematical model. Low reliability and the absence of mathematical models directly affect validity of these objective methods. The measured characteristics of muscle tissue are usually interpreted only based on indirect indicators of their mechanical properties (stiffness, dynamic stiffness, energy dissipation, or curve slope or decay, etc.). Thus, results from different instruments cannot be compared. It is not even possible to compare results measured with one instrument on different muscle groups.

On the other hand, it is evident that there are efforts to use these methods in clinical practice and research (see Section 5). However, in the context of the analyses presented (see Section 4), the question is whether they are being used in a relevant way.

## 7. Conclusions

In this critical review, inconsistencies in the definition of muscle tone were presented, and muscle tone modalities that can be objectively measured were described. On the basis of a comparative analysis, the reliability of objective diagnostic methods for determining mechanical properties of skeletal muscles in vivo, in situ was assessed. Furthermore, clinical practice and research areas in which these methods can be used were outlined.

Based on the findings collected, it can be concluded that the objective assessment of muscle tone can be a valuable, informative tool in diagnosis and choice of therapy in the future. However, the available technologies must be understood as support service tools that cannot and must not replace human judgement. We believe that in further research, two issues need to be addressed. The first of them is the high variance in the reliability of these methods. In the second case, it would be advisable to supplement objective methods of muscle tone diagnosis with mathematical models that would give a clear physical meaning to the observed mechanical characteristics and thus allow for the intercomparison of results.

## Figures and Tables

**Figure 1 sensors-23-07189-f001:**
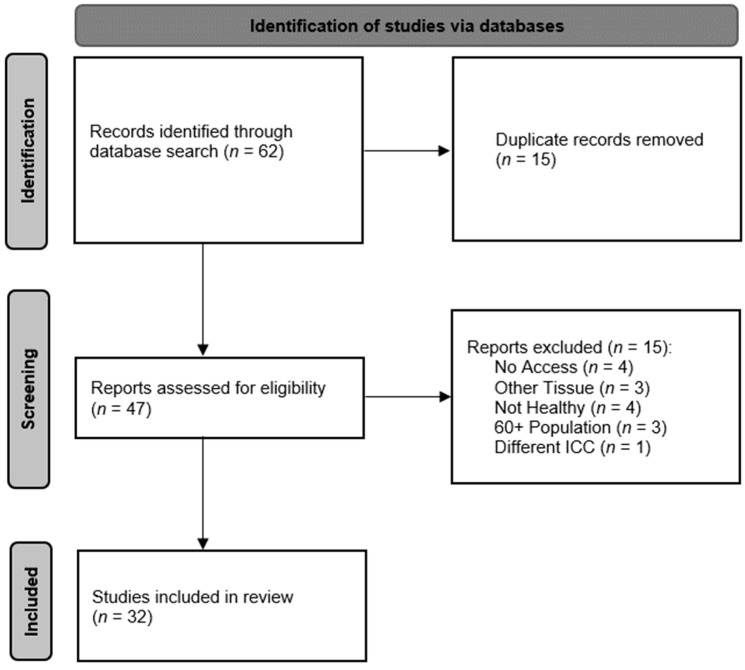
Simplified PRISMA diagram [22] of reliability review process.

## Data Availability

The data presented in this study are available in the relevant referenced studies.

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
