# Peer review of "Objective Methods of Muscle Tone Diagnosis and Their Application—A Critical Review"

_sensors, 2023, doi:10.3390/s23167189_

Round 1

Reviewer 1 Report

No major comments except that introduction is too long, and not focused on the study objectives.

The tense is not matching at places. Some ultra-short sentences have been used, which can gainfully be combined with previous sentences using: therefore, although, however etc.

Author Response

Thank you for your review and insightful comments. The text was linguistically revised again and several edits were made to the tenses and the length of the sentences. Moreover, we added some clarifications. Regarding the length of the introduction, we believe that outlining the palpation techniques is important for the context of the manuscript.

Reviewer 2 Report

The manuscript provides a review of the objective measurement approaches of muscle tone. It is interesting to read, yet it needs a bit more transparency regarding review processes, and a clearer goal of the review.  A few specific concerns are following.

- It seems unclear if the current review answers "How can these methods support subjective palpation examinations?". Maybe, it's a matter of writing.

- In Section 2, the definition of muscle tone was provided. Yet, it would be more informative to provide classical definitions by Lance (1980), Stedman (1982), Foster (1892), Davidoff (1992), and/or Thewlis (1979) first. And also it would be nicer to indicate different components of muscle tone early in Section 2. And then, inconsistencies in operational definitions can be discussed. The current Section 2 seems to be just quick to highlight inconsistencies. Currently, Section 2 does not provide a good overview.

  Foster M. A. (1892). Text Book of Physiology. The Central Nervous System (6th ed.). London: Macmillan, part III

  Stedman T. (1982). Stedman's Medical Dictionary (24th ed.), Williams & Wilkins, Baltimore

  Lance J.W. (1980)The control of muscle tone, reflexes, and movement: Robert Wartenberg Lecture Neurology, 30, pp. 1303-1313

  Davidoff R.A. (1992) Skeletal muscle tone and the misunderstood stretch reflex. Neurology 42:951–963

  Thewlis J. (1979). Concise dictionary of physics and related topics, 2nd edn. Pergamon, Oxford

- In Section 3 (method), provide a more detail of search and review processes (e.g., how many were initially identified, how to decide which one was included or not, whether quality of published work was checked or not, etc.). Table 1 reports ICC values, and I am wondering if the authors specifically selected papers reproing these values.

- In Section 4, Line 173: "this method has proven inadequate." Please provide evidence. And the EMG-based approach seems used in a recent literature (e.g., Kaminishi et al., 2021). 

  Kaminishi, K., Chiba, R., Takakusaki, K., & Ota, J. (2021). Increase in muscle tone promotes the use of ankle strategies during perturbed stance. Gait & Posture, 90, 67–72. 

- SSI and EMG are often used along with a dynamometer. I am not sure if this is properly discussed.

- Table 1. Provide citation information for the included papers and more details about the included papers (e.g., number of participants, population characteristics such as age group, presence of disease).

- Given Section 5, the authors need to indicate this is part of review purposes in the introduction

- The discussion section needs more depth in general.

Author Response

Thank you for your review and insightful comments.

The question “How can these methods support subjective palpation examinations?” is answered throughout the article. The disadvantages of subjective muscle tone examination are described in the last paragraph of Section 2. The advantages as well as shortcomings of the objective methods are described in the subsequent chapters and summarized in Discussion (2nd paragraph). To add some clarity, we added an explicit statement to the end of this paragraph and an introduction to Section 5.

In Section 2, we added the citations of Foster (1892) and Bernstein (2014) and we rearranged the section to first describe the components and then discuss the inconsistencies in definitions, as suggested. The objective of the Section is to illustrate how inconsistent the approach to muscle tone is among different authors and how ambiguous is the term (to support this claim, we added more references - Ganguly 2021, Shortland 2018) and it serves as a basis for the classification (categories) of objective methods for the assessment of muscle tone. To clarify this, we added a paragraph at the end of the Section.

Section 3 was extensively edited and corresponding edits were made to Section 4 as well as Table 1. We chose the ICC index because it is commonly used to assess consistency or reproducibility of quantitative measurements conducted by multiple observers. Given the objective of the paper, we find this parameter to be a suitable indicator of reliability.

In Section 4, we added an explanation and references regarding the inadequacy of EMG.

“SSI and EMG are often used along with a dynamometer. I am not sure if this is properly discussed” - Are you referring to the fact that validity of methods for muscle tone objectification is often tested using dynamometry? Could you please specify what should be discussed and why?

In Table 1, a list of included studies was added. More details on the included papers can be found in Section 3.

We complemented the review objectives outlined in the Introduction to reflect the objectives of Section 5.

Regarding the discussion section, we hope that after the extensive edits of the previous sections, the discussion is more comprehensible and justified now. Could you please be more specific and clarify what should go more in depth?

Reviewer 3 Report

The methods section does not provide specific details on the search terms used, the inclusion and exclusion criteria, or the search strategy employed. It would be beneficial to provide a more detailed explanation of how the search was conducted to ensure transparency and reproducibility.

A more comprehensive search strategy, including multiple databases and considering studies in different languages, would enhance the reliability and representativeness of the findings.

The methods section does not provide details on how the studies were selected for inclusion in the review. It is important to describe the criteria used for study selection, the process of screening and assessing the eligibility of studies, and any conflicts or disagreements resolved during the selection process. This information is crucial to ensure transparency and minimize potential bias.

The methods section does not mention how data extraction was performed or how the findings from the selected studies were synthesized. It is important to describe the process of extracting relevant data from the included studies and how the data were analyzed and synthesized to provide a comprehensive overview of the available objective methods for the diagnosis of muscle tone.

The section lacks specific information on how the objective methods were applied in physiotherapy, ergonomics, and sports. It would be helpful to provide more details on the procedures followed, including the equipment used, measurement protocols, and data analysis methods.

The section primarily focuses on the use of Myoton for muscle tone assessment, with minimal mention of other devices or techniques. Including a broader range of objective methods and comparing their strengths and limitations would provide a more comprehensive analysis.

The section does not address the limitations or potential challenges associated with using objective methods for muscle tone assessment. It is important to acknowledge any limitations in accuracy, reliability, or applicability of these methods in different contexts.

Are there any potential limitations or challenges associated with using objective methods for muscle tone assessment in these fields? How do these limitations impact the validity and reliability of the measurements?

It would be beneficial to include studies or research findings that support the claims made regarding the limitations and advantages of palpation versus instrumental methods.

What empirical evidence supports the argument that palpation cannot be replaced by instrumental methods? Are there any comparative studies or clinical trials that demonstrate the superiority of palpation in certain contexts?

How can the high variability of reliability in instrumental methods be addressed? Are there specific factors or variables that contribute to this variability, and can they be controlled or minimized?

What are the specific scenarios or clinical situations where instrumental methods provide valuable information that palpation alone cannot capture? Are there any examples or case studies that highlight the advantages of instrumental methods in diagnosing muscle tone and guiding therapy choices?

Need to be checked again after final version is approved. 

Author Response

Finding many of the comments unreasonable (asking for information that is already included in the manuscript and for additions that would make the manuscript extremely long and not focused on its objectives, or suggesting reviewing studies in different languages), we decided to use some engines to check whether the text was written by AI. Both AI Detector and UNDETECTABLE.AI confirmed that this Review Report was most likely written by AI. We decided to raise this issue to the editor and addressed the comments as such.

The comment about the methods section is reasonable to some extent and we made extensive edits along with corresponding edits to Section 4 and Table 1. However, we would like to point out that all four paragraphs suggesting revision say pretty much the same in different words (i.e., we believe that there is no need to repeat four times that inclusion criteria have to be detailed). English is the principal academic language and we find the suggestion to use other languages quite unreasonable.

Regarding Section 5, we would like to point out that the purpose of this section is to provide a selection of research areas in which the objective methods of muscle tone can be used. Citing 60+ studies, providing more details on the procedures followed would make the manuscript excessively long and distract from the purpose of the section. The risks and shortcoming of these methods are collectively described in the previous section. To add some clarity, we added an introduction to Section 5.

In Paragraph 6, Reviewer 3 does not even specify which section he refers to. In Section 5, Myoton is mentioned most frequently because it is most frequently used, probably since it is an affordable and easy-to-use device compared to other methods (see also the last paragraph of the sports section). However, the section does not focus on specific methods but on their use in general, and other methods are mentioned as well.

We believe that limitations and potential challenges associated with using the objective methods are sufficiently described in Sections 4 and 6.

We also believe that the advantage of objective methods over subjective methods is obvious from the nature of the words. We added an introduction to Section 5 to increase clarity. The opinion that palpation cannot be replaced by instrumental methods is referenced (Kolar 2014, Lewit 2009). The shortcomings of instrumental methods are described as well, especially in terms of reliability.

Regarding the variability of the reliability, we added an explicit sentence to the discussion section.

Regarding the last paragraph, we believe that it was addressed by the replies above (subjective vs. objective methods, introduction to Section 5).

Round 2

Reviewer 2 Report

I thank the authors for the revised version. 

In the earlier review, I had a concern about the discussion, though I didn't provide much specific suggestions.  I understand the overall conclusion. Yet, given the limitations of the objective methods (i.e., high variability, no model), the conclusion is not so convincing – "the objective assessment of muscle tone can be a valuable, informative tool in diagnosis and choice of therapy." The objective methods certainly have a potential, though it sounds like more work should be done. I wished that the discussion was developed enough to clearly support the conclusion. 

I recommend minor revision, though highly encourage the authors to improve the conclusion. I've shared this encouragement with the editors, so it can be addressed.

Author Response

Thank you for your insightful comments again.

We divided Section 6 into Discussion and Conclusion and both sections were expanded. Hopefully, now it is more clear from the last section that more work should be done around the objective methods so that they could be used reliably and widely.

Reviewer 3 Report

Thank you for modifying your manuscript following the comments and suggestions. 

Author Response

Thank you for your approval.